# Functional properties of stellate cells in medial entorhinal cortex layer II

**David C Rowland\*, Horst A Obenhaus, Emilie R Skytøen, Qiangwei Zhang, Cliff G Kentros, Edvard I Moser, May-Britt Moser\***

Kavli Institute for Systems Neuroscience and Centre for Neural Computation, Norwegian University of Science and Technology, Trondheim, Norway

**Abstract** Layer II of the medial entorhinal cortex (MEC) contains two principal cell types: pyramidal cells and stellate cells. Accumulating evidence suggests that these two cell types have distinct molecular profiles, physiological properties, and connectivity. The observations hint at a fundamental functional difference between the two cell populations but conclusions have been mixed. Here, we used a tTA-based transgenic mouse line to drive expression of ArchT, an optogenetic silencer, specifically in stellate cells. We were able to optogenetically identify stellate cells and characterize their firing properties in freely moving mice. The stellate cell population included cells from a range of functional cell classes. Roughly one in four of the tagged cells were grid cells, suggesting that stellate cells contribute not only to path-integration-based representation of self-location but also have other functions. The data support observations suggesting that grid cells are not the sole determinant of place cell firing.
DOI: https://doi.org/10.7554/eLife.36664.001

## Introduction

The medial entorhinal cortex (MEC) is thought to create a map of space through a set of functionally distinct cell types: grid cells, border cells, head direction cells, and speed cells (*Rowland et al., 2016*). Each functional cell type follows its own developmental trajectory (*Bjerknes et al., 2014*; *Langston et al., 2010*; *Wills et al., 2010*), suggesting that the functional identity of the cell is hardwired during development. A reasonable hypothesis is therefore that these functional cell types map onto the diversity of morphologically or molecularly defined cell types found in the MEC. Layer II of the MEC contains two largely distinct populations of principal cells: stellate cells and pyramidal cells. Stellate cells express reelin (*Pesold et al., 1998*; *Kitamura et al., 2014*; *Fuchs et al., 2016*; *Winterer et al., 2017*), have large sag potentials (*Dickson et al., 2000*), superficially branching dendrites (*Canto and Witter, 2012*), and, at least in rodents, sub-threshold membrane potential oscillations in the theta range (*Alonso and Llinás, 1989*). Pyramidal cells form tight clusters (*Kitamura et al., 2014*; *Ray et al., 2014*), express calbindin and WFS-1 (*Kitamura et al., 2014*), have thick apical dendrites (*Canto and Witter, 2012*), and receive strong cholinergic input from the medial septum (*Ray et al., 2014*). Within the local circuit, the two-cell populations form largely distinct microcircuits: pyramidal cells are connected via 5-HT 3a positive interneurons (*Fuchs et al., 2016*), while stellate cells are connected via fast-spiking parvalbumin (PV+) cells and slower spiking somatostatin-positive cells (*Fuchs et al., 2016*). However, the two subnetworks are not entirely independent; it has been estimated that up to 14% of pyramidal cells project to stellate cells (*Winterer et al., 2017*) and some cells express an intermediate stellate/pyramidal identity (*Fuchs et al., 2016*).

The functional identity of stellate cells and pyramidal cells has implications for the broader hippocampal-entorhinal circuit. Nearly every stellate cell projects to the DG, CA3 and/or CA2 regions of the hippocampus, where terminals from stellate cells make up the main, and nearly exclusive,

**\*For correspondence:**
david.c.rowland@ntnu.no (DCR);
may-britt.moser@ntnu.no (M-BM)

**Competing interests:** The authors declare that no competing interests exist.

excitatory input from the MEC (*Kitamura et al., 2014*; *Ray et al., 2014*; *Varga et al., 2010*). The targets of layer II pyramidal cells remain largely unclear. A subset of the cells project to the CA1 region (*Kitamura et al., 2014*) and to the contralateral hemisphere of the MEC (*Varga et al., 2010*), but importantly, they do not terminate in the DG, CA3 or CA2 regions (*Kitamura et al., 2014*; *Ray et al., 2014*). Because some theoretical models propose that the spatial specificity of cells in the DG and CA fields are formed in large part by combining input from grid cells (*Solstad et al., 2006*; *McNaughton et al., 2006*; *Savelli and Knierim, 2010*; *Monaco et al., 2011*; *de Almeida et al., 2012*), determining the functional identity of the two MEC layer II populations is critically important.

Previous attempts to functionally characterize stellate cells have produced mixed results. Intracellular recordings of stellate and pyramidal cells in mice running on a virtual linear track have shown that both types could be putative grid cells (*Domnisoru et al., 2013*; *Schmidt-Hieber and Häusser, 2013*) and imaging studies of freely moving animals have found the same (*Sun et al., 2015*). These findings contrast with those of another study, which assigned cells into putative stellate or pyramidal cells based on their phase locking to the ongoing theta oscillation. Based on this classifier, only 3 of 94 putative stellate cells (near chance levels) were grid cells (*Tang et al., 2014*), calling into question the role of grid cells in enabling spatial firing in target regions of stellate cells, such as the hippocampus. Here, we attempt to reconcile these differences using a mouse line in which the inhibitory opsin ArchT (*Chow et al., 2010*; *Han et al., 2011*) is expressed almost exclusively in stellate cells. Using ArchT expression and optogenetic inhibition to identify stellate cells, we first showed that approximately one in four stellate cells are grid cells. We then revisited the previously published classifier using an extended dataset of over 1300 cells and 400 grid cells recorded in layer II. Consistent with prior observations (*Latuske et al., 2015*), we found that neither the classifier's cell assignment nor the relationship between the firing of the cell and the theta oscillation cleanly separated grid cells from other cell types. Taken together, the results suggest that substantial fraction of the stellate cells are grid cells, although they can have other functional identities as well, and the relationship between the cell's firing and the theta oscillation is a poor predictor of the functional identity of the cell.

## Results

We sought to characterize the functional properties of stellate cells using an optogenetic tagging approach. To gain genetic access to stellate cells, we used the 'EC-tTA' line (*Yasuda and Mayford, 2006*) that expresses almost exclusively in the parahippocampal region (entorhinal cortex together with the pre- and para- subiculum). Within the MEC, where we targeted our tetrodes, the expression is confined to layer II (*Figure 1*). Previous work on the 'EC-tTA' line has shown that the transgene-expressing cells in layer II of the MEC have stellate-like morphologies, project to the dentate gyrus and CA3 regions of the hippocampus (*Rowland et al., 2013*), and often show distinctive circular gaps in expression (*Kanter et al., 2017*), which presumably correlate with circular clusters of calbindin-positive pyramidal cells (*Kitamura et al., 2014*) and would be expected if the transgene-expressing cells were stellate cells. Because the tTA-line allows expression of our transgene of interest in stellate cells, we crossed the line to a tetO-ArchT-GFP (*Weible et al., 2014*) line to enable functional characterization of these cells through optogenetic tagging. To characterize expression of ArchT in this particular cross, we combined an in situ hybridization staining for the transgene with antibody staining for either calbindin, a marker for pyramidal cells (*Figure 1A,B*), or in situ staining for reelin, a marker for stellate cells (*Figure 1C,D*), on alternating sections of one mouse. Arch-expressing cells avoided clusters of calbindin-positive cells (*Figure 1B*) but overlapped extremely well with reelin (*Figure 1D*). Overall, only a small minority of the Arch-expressing MEC cells expressed calbindin (3 of 303 Arch-expressing cells expressed calbindin, ~ 1%; 3 of 433 calbindin expressing cells expressed Arch, less than 1%) while 97.2% expressed reelin (315 of 324 ArchT-expressing cells expressed reelin; 315 of 967 of reelin expressing cells expressed ArchT, 32.5%; *Figure 1C,D*). The fact that not all stellate cells express Arch raises the possibility that the EC-tTA line targets a subclass of stellate cells, but we could not test for that because there are no known markers for any subclass. To check for consistency across animals, we also examined the overlap between ArchT and reelin in three additional mice and each additional mouse showed a very similar pattern (*Figure 1—figure supplement 1*). Finally, to confirm that the ArchT-expressing cells project to the hippocampus, we checked for GFP-labeled fibers in the molecular layer of the dentate gyrus and CA3, the expected

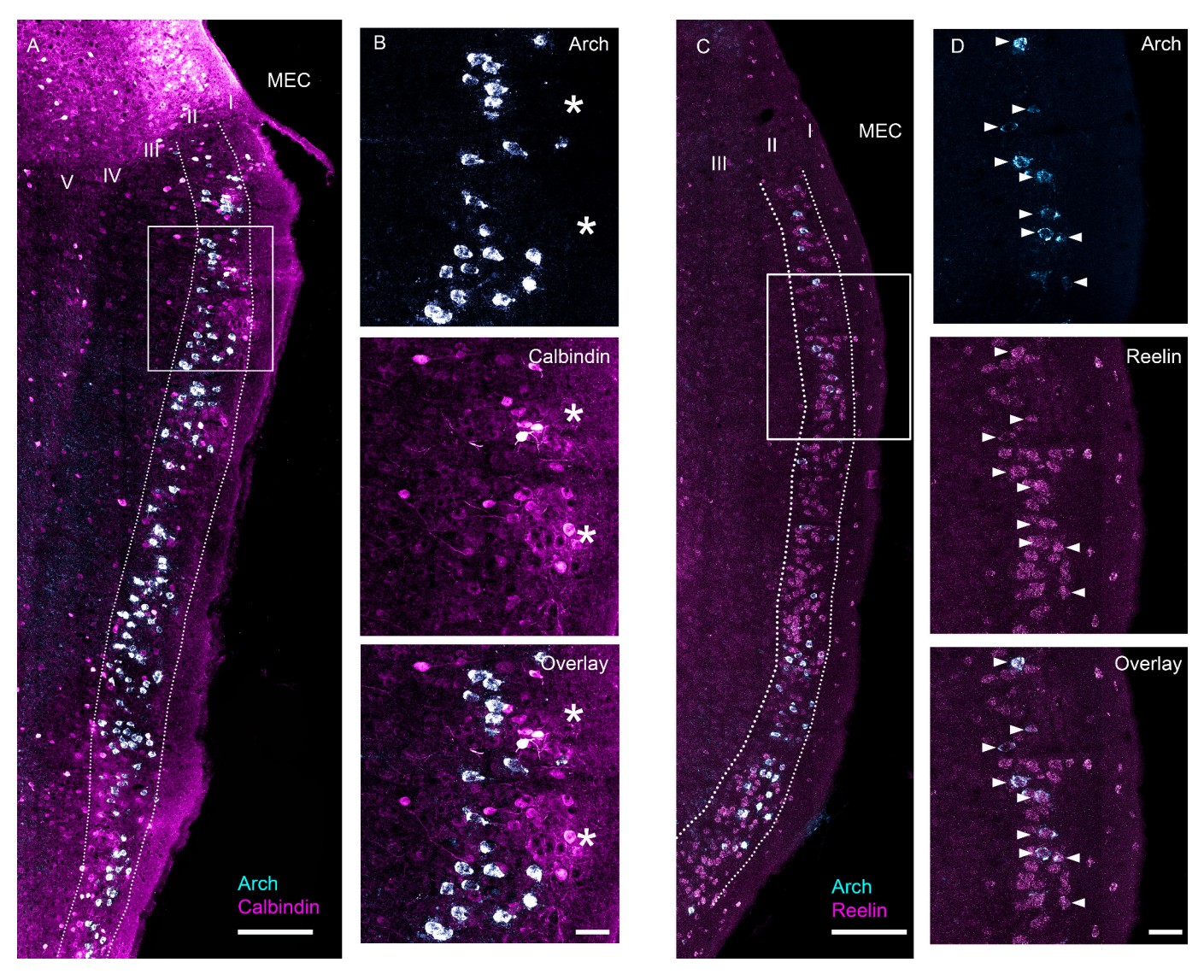

**Figure 1.** Characterization of ArchT expression in the EC-tTA x tetO-ArchT-GFP transgenic mouse line. Images are 63x scans acquired under a confocal microscope at multiple focal planes and collapsed into a single maximum intensity projection. (**A**) Overlap between Arch mRNA expression (cyan) and calbindin, a marker for MEC layer II pyramidal neurons, protein expression (magenta). (**B**) Blowup of boxed area in (**A**) for Arch mRNA expression (top), calbindin protein expression (middle) and overlay (bottom). Asterisks indicate islands of calbindin-positive cells. None of the Arch-expressing cells in this section also expressed calbindin. (**C**) Overlap between Arch mRNA expression (cyan) and reelin, a marker for MEC layer II stellate cells, mRNA expression (magenta). (**D**) Blowup of boxed area in (**C**). All the Arch-expressing cells also expressed reelin in this field of view. Scale bars are 200 um for overview images and 50 µm for blowups.

DOI: https://doi.org/10.7554/eLife.36664.002

The following figure supplements are available for figure 1:

**Figure supplement 1.** Examples of ArchT and reelin mRNA labeling in three additional EC-tTA x tetO-ArchT-GFP mice.
DOI: https://doi.org/10.7554/eLife.36664.003

**Figure supplement 2.** Arch-expressing cells project to the dentate gyrus (DG) and CA3 regions of the hippocampus.
DOI: https://doi.org/10.7554/eLife.36664.004

**Figure supplement 3.** Nissl-stained saggital brain sections showing recording locations for seven EC-tTA x tetO-ArchT-GFP mice used in the study and three additional wild-type animals used in the extended layer II dataset.
DOI: https://doi.org/10.7554/eLife.36664.005

termination zones of MEC stellate cells, and found intense labeling in these areas (*Figure 1—figure supplement 2*). The cross therefore drives expression of ArchT almost exclusively in stellate cells in MECII.

To functionally characterize the transgene-expressing cells, we implanted a bundle of four tetrodes glued to an optic fiber (see Materials and methods) into the MEC (*Figure 1—figure supplement 3*) of seven double positive EC-tTA x tetO-ArchT-GFP mice. After recovery from surgery, single units were recorded as the mice foraged for food in a 0.8 or 1 m wide square open field (*Figure 2A*). Immediately following the open-field session, the mice were placed in a smaller box (27cm × 20 cm×14 cm height) and connected to a 532 nm green laser via a patch cable for an optogenetic tagging session. Optogenetic tagging is typically performed using excitatory opsins (*Lima et al., 2009*; *Zhang et al., 2013*). However, inhibition has been used for the same purpose (*Wolff et al., 2014*) and has the advantage that it circumvents the problem of false positives caused by recurrent excitation. Previous work has shown that Arch silences cells with near-zero latency (*Chow et al., 2010*; *Wolff et al., 2014*), and we therefore reasoned that we could reliably tag ArchT-expressing cells using short (10 to 20 ms) pulses of light delivered thousands of times at 1–4 Hz (*Figure 2B,C*; additional examples in *Figure 2—figure supplement 1*). The large number of trials was necessary to have sufficient data for reliable statistics and to accurately estimate the latency to inhibition because the cell needs to be active precisely when the light comes on. We spike-sorted the baseline and tagging sessions together to avoid cluster assignment errors, and only analyzed cells with fewer than 1% of spikes between 0 and 2 ms in the interspike interval histogram and with high waveform

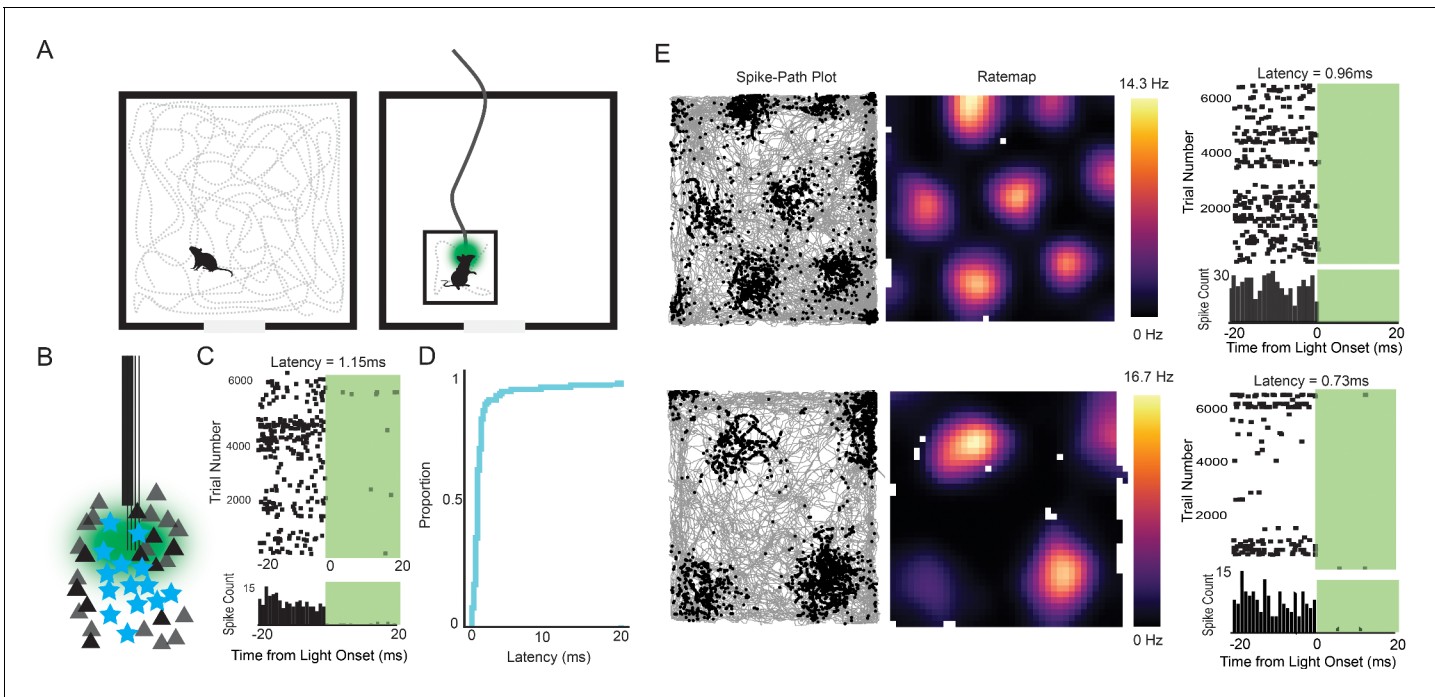

**Figure 2.** Experimental design and examples of tagged grid cells. (**A**) Mice were run in a 0.8 or 1 m box for 30–40 min while cells in the MEC were recorded. The mice were then connected to a patch cable and placed in a towel-lined holding box for an optogenetic tagging session. The tagging session consisted of 10–20 ms light pulses delivered thousands of times at 1–4 Hz. (**B**) Cartoon of the experimental logic. ArchT-expressing stellate cells (cyan stars) should be immediately inactivated by light delivery while neighboring pyramidal cells (black triangles) should not, allowing functional identification of stellate cells. (**C**) Raster plot (top) and histogram (bottom; bin size = 1 ms) for an example tagged cell. The cell was silenced with an estimated latency of 1.15 ms. (**D**) Cumulative histogram of latencies for all inhibited cells. Median latency was 0.85 ms. (**E**) Two further example tagged grid cells (spike-path plots on left and color-coded rate maps with colorbar in the middle) and their corresponding raster plots and histograms from the tagging session (right). Both cells had sub-millisecond latencies.

DOI: https://doi.org/10.7554/eLife.36664.006

The following figure supplement is available for figure 2:

**Figure supplement 1.** Four additional tagged cells.

DOI: https://doi.org/10.7554/eLife.36664.007

correlations (r > 0.99) between the baseline and tagging session. We considered the cell to be silenced if the activity in the baseline period was more than during the laser-on period (two sided t-test with stringent cutoff of p<0.001). The latency to inhibition was determined using a change point analysis (*Wolff et al., 2014*). Most of the cells had near-zero millisecond latencies (median latency = 0.85 ms, *Figure 2C–E*), making it highly improbable that silencing was caused through a network mechanism. We applied a latency threshold of 5 ms, which eliminated three outlier cells and left a total of 75 tagged cells from a population of 578 (~13% tagged cells). The non-tagged population included non-expressing or non-responsive stellate cells as well as all other cell types (see below).

To assign the cells into functional classes, we computed standard scores for speed modulation, head directionality, gridness, and border-related firing (see Materials and methods). We then shuffled the spike times of each cell 200 times, recalculated the measures each time, and used the 95th percentile of the shuffled distribution as a cutoff for assigning cells to a particular class. For head direction cells, we additionally required that the correlation of the directional firing of the first half and the second half of the trial exceed 0.6. Cells that passed more than one criterion were considered mixed (light blue portion of bars in *Figure 3A*) and counted in all supra-threshold categories (thus, the sum exceeds 100%). The tagged population included grid cells (25.3%; 16% pure, 9.3% mixed), head direction cells (12%; 9.3% pure, 2.7% mixed), speed cells (16%; 8% pure, 8% mixed), border cells (1.3%; 1.3% pure, 0% mixed), and unclassified cells (56.0%). To control for the possibility that accidental resampling of cells between sessions created spurious results, we downsampled our data to cells recorded 4 or more days apart, when the electrodes had been turned approximately 100 microns. The downsampled population had similar proportions in each functional class as the full dataset, suggesting that percentages described here are representative of the functional diversity in layer II of the MEC (*Figure 3—figure supplement 1*). We next compared the properties of the tagged cells to the untagged population. The untagged population likely included layer II pyramidal cells, non-expressing stellates (i.e reelin-expressing cells that do not express Arch, *Figure 1C, D*) and possibly even some layer III cells that fell within the ~100 micron recordable distance of the tetrodes (*Gray et al., 1995*; *Harris et al., 2000*). The untagged population therefore represents the properties of superficial layer cells of multiple morphologically- and molecularly- defined cell types. The untagged population included grid cells (35.6%; 26.8% pure, 8.8% mixed), head direction cells (9.1%; 6.1% pure, 3% mixed), speed cells (18.9%; 10.1% pure, 8.8% mixed), border cells (5.8%; 4.6% pure and 1.2% mixed), and unclassified cells (41.6%). We next compared the raw scores between

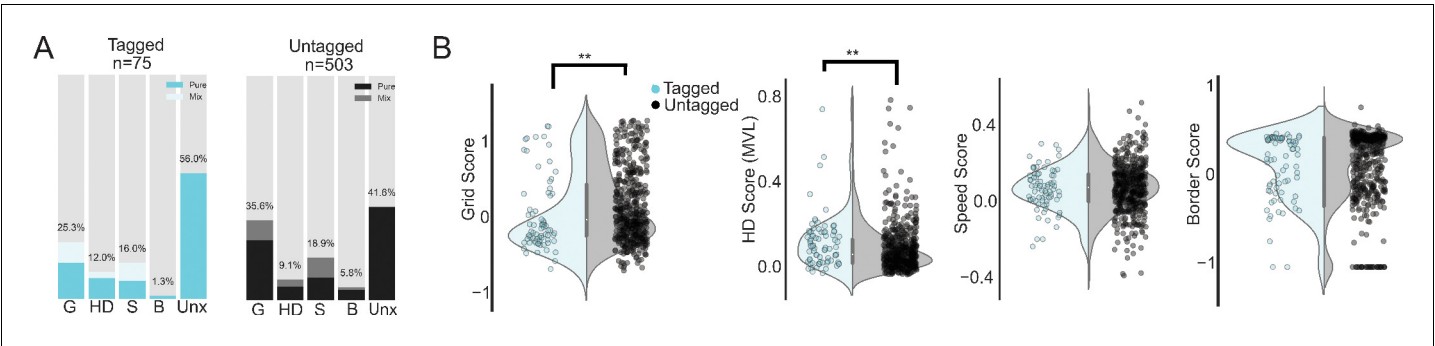

**Figure 3.** Properties of tagged cells. (**A**) Percentage of tagged (cyan, left) and untagged (black, right) cells in the unclassified category (Unx) and each of the 4 functional cell classes: grid (G), head direction (HD), speed (S) and border cells (B). Light regions of the bar plots show the percentage of cells in each class that belonged to more than one class ('Mix'). Note that because a cell could belong to more than one class, the total percentages exceed 100. (**B**) Violin plots of grid, head direction, speed and border scores where the shaded region gives kernel density estimate for tagged (light cyan) and untagged cells (grey). Individual data points overlaid on top of the violins. Asterisks indicate a significant difference between the two populations (**p<0.01, all tests Mann-Whitney U-tests).

DOI: https://doi.org/10.7554/eLife.36664.008

The following figure supplement is available for figure 3:

**Figure supplement 1.** Distribution of functional cell types for the whole dataset (**A**) and a downsampled dataset of all cells with sessions separated by 4 or more days (**B**), when the electrodes had been turned 100 microns.

DOI: https://doi.org/10.7554/eLife.36664.009

the populations using two-sided, non-parametric statistics (*Figure 3B*). The untagged population had lower head direction scores (two-sided Mann-Whitney U test, U = 22869.0, p=0.003), higher grid scores (Mann-Whitney U test, U = 14225.0, p=0.0037), and lower spatial information content (two-sided Mann-Whitney U test, U = 21692.0 p=0.036) compared to the tagged cells. There was no significant difference in border score (two-sided Mann-Whitney U test, U = 16536.0, p=0.085), or speed score (two-sided Mann-Whitney U test, U = 18307.0, p=0.68). We also tested whether the grid scores of cells categorized as grid cells (two-sided Mann-Whitney U test, U = 1786.0, p=0.72), head direction score of head direction cells (two-sided Mann-Whitney U test, U = 208.0, p=0.99), and speed score of speed cells (two-sided Mann-Whitney U test, U = 461.0, p=0.28) differed between the tagged and untagged populations, and we found no significant differences. The number of border cells was too small to do a statistical comparison. In summary, the tagged and untagged populations were heterogeneous but notably both groups included substantial numbers of grid cells (approximately 1 in 4 for tagged cells and 1 in 3 for untagged cells).

These results contrast with a previous study by Tang and colleagues (*Tang et al., 2014*). Tang et al. assigned extracellularly recorded cells into putative pyramidal and stellate classes based on depth of modulation and preferred firing phase relative to the ongoing theta oscillation. The authors concluded that stellate cells were grid cells only ~3% of the time (near chance levels) but were more frequently border cells (~11%). In contrast, putative pyramidal cells were more often grid cells (~19%) and less often border cells (~1%). We first asked how well the classifier approach worked to identify our tagged cells. We found that the classifier identified our tagged cells as putative stellate cells 81.3% of the time (*Figure 4A*). Most of the misidentified cells were clearly in the putative pyramidal region and the tagged grid cells showed a similar distribution as the other tagged cells (*Figure 4—figure supplement 1*; mean distance to decision boundary for the misidentified cells = 0.87). Because the untagged population also includes stellate cells, among others, we cannot determine the number of cells falsely identified as stellate cells and therefore cannot fully evaluate the classifier. Nevertheless, we note that our calculated true positive rate for stellate cells is only around 80%. Therefore, there is currently no available substitute for identifying stellate cells with high precision besides more labor intensive options like functional imaging combined with a marker of stellate cells (*Sun et al., 2015*), intracellular recordings with post-hoc reconstructions (*Domnisoru et al., 2013*; *Schmidt-Hieber and Häusser, 2013*; *Burgalossi et al., 2011*), or optogenetic tagging, as we have done here.

The Tang et al. results also suggested that theta phase and strength could be used to predict the functional identity of the cells. We therefore evaluated their classifier on an extended dataset of 1332 cells including 411 grid cells. This extended dataset included recordings from the EC-tTA x tetO-ArchT-GFP mice and three additional mice with histologically confirmed layer II recordings (N = 270 putative pyramidal cells, 999 putative stellate and 63 cells in the guard zone). We observed a similar distribution in plots of the preferred firing phase and strength of theta modulation as Tang et al. (*Figure 4B*). The preferred firing phase of the cells was not evenly distributed across the theta cycle (Rayleigh test for non-uniformity, p=$3.79 \times 10^{-9}$) and there were two peaks in the distribution: one near the peak and one near the trough of the oscillation (*Figure 4B*, *Figure 4—figure supplement 2*; see also [*Newman and Hasselmo, 2014*]). In contrast to the Tang et al. findings, however, we found that cells classified as stellate cells had overall higher grid scores than cells that were classified as pyramidal cells (two-sided Mann-Whitney U test, U = 114913, p=0.0003). Correspondingly,~32% of the putative stellates were grid cells, compared to ~23% of the putative pyramidal cells (*Figure 4C,D*). We did not find a significant difference in border scores between the two populations (two-sided Mann-Whitney U-Test, U = 127330.5, p=0.158, *Figure 4C*).

We then considered whether grid cells preferentially fired at a particular phase of the theta oscillation. Individual grid cells showed large diversity in their preferred firing and depth of theta modulation (*Figure 4B,E*) as expected given the propensity of layer II grid cells to phase-precess (*Hafting et al., 2008*). Clustering of the cells using an agglomerative clustering approach captured one population that fired near the peak and one that fired near the trough of the theta oscillation. Similar to what we found when we used the classifier, the clustering approach revealed higher grid scores in the peak-preferring population but cells with high grid scores in both populations (*Figure 4—figure supplement 2*). Border scores were not significantly different between the two populations.

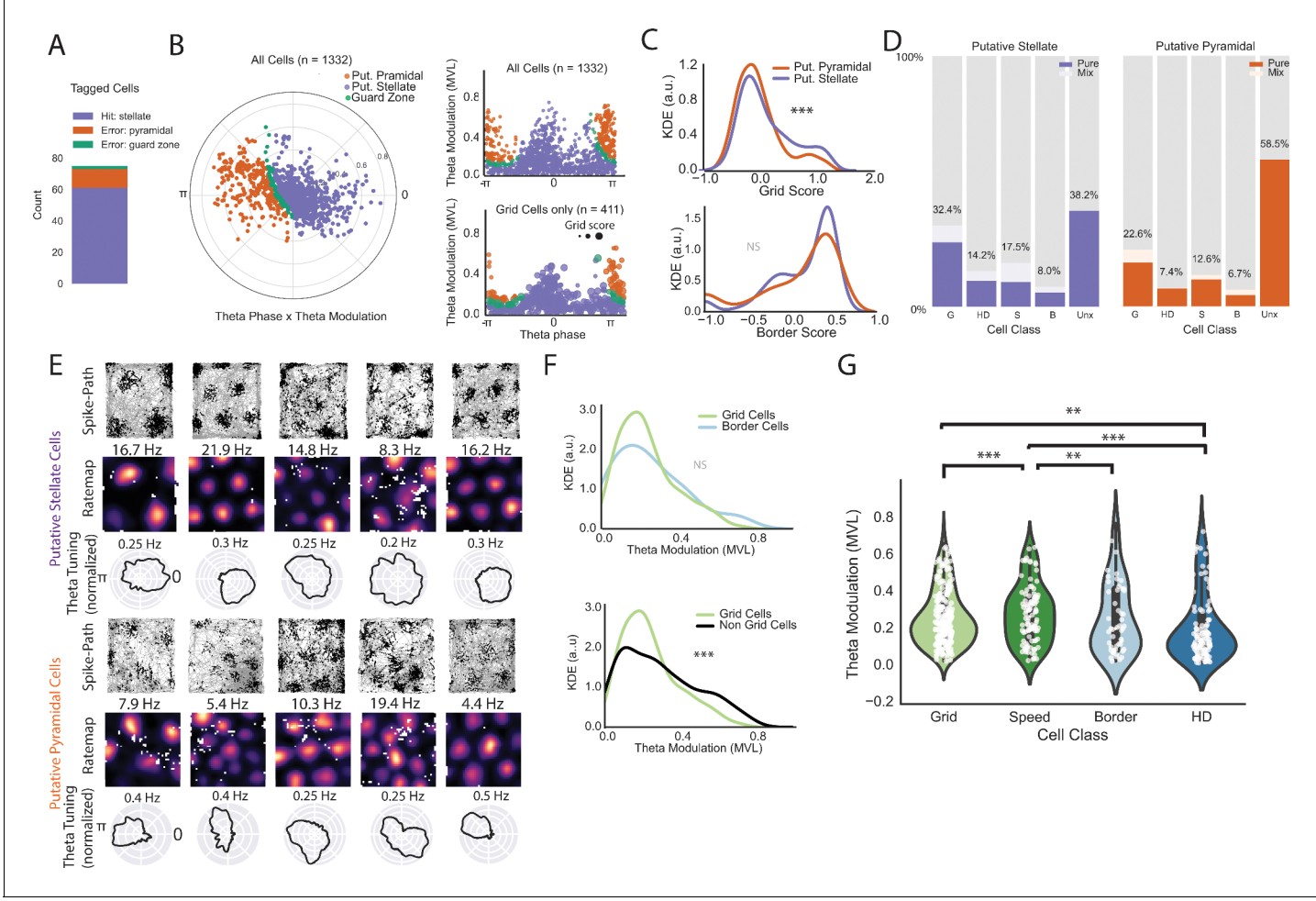

**Figure 4.** Classifier performance and comparison of theta phase locking for different functional classes on an extended layer II dataset. (**A**) Classifier performance on the population of tagged cells. Numbers of cells correctly identified as stellate cells (purple), incorrectly identified as pyramidal cells (orange), and in the guard zone (green) are shown in the stacked bar chart. (**B**) Preferred theta phase (angle) and strength of theta phase modulation (mean vector length, MVL, radius, peak of the oscillation is 0 and trough is pi) for all cells in in the extended layer II dataset. The data for all cells are presented in circular form on the left and unwrapped on the top right. Cells are color-coded based on their classification into putative stellate and pyramidal categories (classification based on phase locking to local theta oscillation; see text). Cells showed some clustering around the peak (0) and trough (pi). Bottom right shows the distribution of grid cells only where the dot size is proportional to the grid score. (**C**) Kernel density estimates (KDE) of grid scores (top) and border scores (bottom) for putative stellate cells and putative pyramidal cells. (**D**) Breakdown of the two groups by functional category. (**E**) Example grid cells (path plots and color-coded rate maps) with theta phase histograms (normalized such that the area under the curve equals 1 for comparison of depth of modulation between cells) for the two categories of cells. Peak rates for rate maps are indicated above the maps. Clear grid cells exist in both populations and exhibit a variety of theta phase preferences. (**F**) KDEs showing that grid cells exhibit no significant difference in theta modulation from border cells (top) but less theta modulation than non-grid cells as a group (bottom). (**G**) Violin plots with individual data points in white overlaid for theta modulation by cell class (shaded regions give the kernel density estimates and the white dots are individual data points). We only included pure cells (cells that classified criteria for only one cell type) in this analysis to preserve independence between groups. (**=P < 0.01, ***=P < 0.001, two-sided Mann-Whitney U-test).

DOI: https://doi.org/10.7554/eLife.36664.010

The following figure supplements are available for figure 4:

**Figure supplement 1.** Performance of the Tang et al
DOI: https://doi.org/10.7554/eLife.36664.011
**Figure supplement 2.** A second approach to clustering the data gives similar results.
DOI: https://doi.org/10.7554/eLife.36664.012
**Figure supplement 3.** Grid score is negatively correlated with depth of theta modulation.
DOI: https://doi.org/10.7554/eLife.36664.013

We then asked whether grid cells had stronger theta modulation than non-grids. As a group, despite having many strongly modulated example cells, grid cells were significantly less theta modulated, as measured by the mean vector length of the cell firing with respect to the theta oscillation, than non-grids (*Figure 4F*; two-sided Mann-Whitney U test, U = 162644.0, p=9.28×10$^{-5}$), perhaps again reflecting the strong phase precession of layer II grid cells (*Hafting et al., 2008*). Consistent with these results, the strength of theta modulation had a very weak but significant negative correlation to grid score (Pearson's r = −0.15, p=6×10$^{-8}$; *Figure 4—figure supplement 3*). We next compared the depth of theta modulation by class. To avoid comparing cells against themselves, we compared only cells that were 'pure' (i.e. above threshold in just one category; number of pure grid cells = 327, pure speed cells = 127, pure border cells = 68, pure head direction cells = 121). Pure grid cells were not significantly different in theta modulation compared to pure border cells (two-sided Mann-Whitney U test, U = 22816, p=0.087, *Figure 4G*), but had less theta modulation than pure speed cells (two-sided Mann-Whitney U test, U = 16548.0, p=0.0002) and more theta modulation than pure head direction cells (two-sided Mann-Whitney U test, U = 23391.0, p=0.003). In summary, theta modulation is prevalent across all functional cell types in the MEC but with some small, although significant, differences between groups. For example, we observed a small negative correlation between grid score and depth of theta modulation but the correlation accounted for approximately 2% of the variance. These results show that, despite some differences between groups, the large diversity in theta modulation and preferred theta phase within cell types makes the relationship between the firing of the cell and the theta oscillation a poor predictor of the functional identity of the cell, in close agreement with the results of *Latuske et al., 2015*.

## Discussion

We have shown that stellate cells in layer II of the MEC can belong to multiple functional cell classes and that approximately one in four are grid cells. These results serve to reconcile conflicting evidence in the field. On the one hand, in vivo whole cell recordings and functional imaging of stellate cells have both found approximately equal or greater numbers of putative grid cells in the stellate population than in the pyramidal cell population (*Domnisoru et al., 2013*; *Schmidt-Hieber and Häusser, 2013*; *Sun et al., 2015*). On the other hand, it has been reported, in work where cell identity was classified on the basis of locking to the theta phase, that putative stellates are almost never grid cells (*Tang et al., 2014*). We addressed this controversy in two ways. First, we optogenetically tagged stellate cells using a mouse line that expresses transgenes almost exclusively in stellate cells of MEC layer II. Second, we used the previously published classifier *(Tang et al., 2014)* on a dataset of 1332 cells and 411 grid cells.

Optogenetic tagging showed that roughly 25% of stellate cells are grid cells. The actual proportion of grid cells may be larger than this estimate, for three reasons: (i) spike sorting errors can lead to misassignment of cells into the wrong functional class (*Navratilova et al., 2016*), (ii) boundary-induced distortions of the grid pattern could, in theory, cause a grid cell to fall below the cutoff (*Stensola et al., 2015*), and (iii) grid cells with inter-field spacing larger than our 0.8–1.0 meter recording box would be missed in our analysis (*Stensola et al., 2012*). The widespread presence of grid cells in the stellate cell population was corroborated by the use of the theta classifier, which showed that putative stellate cells are more likely to be grid cells than putative pyramidal cells, in agreement with a previous study re-testing the same classifier (*Latuske et al., 2015*). The classifier results should be taken with a note of caution, because our data and the Tang et al. data suggest that the classifier correctly assigns stellate cells only around 80% of the time. One unanswered question is why the classifier assigned grid cells almost exclusively to the putative pyramidal category in the original Tang et al. study, but not in the present study and the Latuske et al. study. One clear difference is that the number of cells and animals recorded in the present study and the Latuske et al. study is higher than in the Tang et al. study. A second difference is the fraction of grid cells reported in the studies. We report approximately 30% grid cells in our layer II recordings using the 95th percentile cutoff and the Latuske study reported around 20% using the 99th percentile cutoff. In contrast, the Tang et al. study reported only 10% grid cells at the 95th percentile cutoff, well below any other published estimate with layer II tetrode recordings in rodents. As there are now clear indications from imaging work that grid cells are not homogeneously distributed in layer II (*Heys et al., 2014*), one possibility is that Tang et al. did not sample from a large enough region of layer II to get

a representative sample. However, the lack of histology in the Tang et al. paper makes it impossible to compare recording locations between the studies.

Although the suggestion that one of four stellate cells probably is an underestimate, the majority of stellate cells are probably not grid cells. The remaining cells include smaller fractions of cells of all functional types, as well as a large number of unclassified cells, which might include the recently discovered object vector cells (*Hoydal et al., 2018*). This is consistent with recent observations of the role of PV+ cells, which contact stellate cells but not pyramidal cells (*Fuchs et al., 2016*). On the one hand, PV+ cells are preferentially connected with grid cells in vivo (*Buetfering et al., 2014*) and manipulations of the PV+ network alters grid cell firing (*Fuchs et al., 2016*). On the other hand, the effects of manipulating the PV+ network are not strictly limited to grid cells alone (*Buetfering et al., 2014*; *Miao et al., 2017*). Moreover, although we cannot address the functional identity of layer II pyramidal cells using our approach due to the high numbers of non-expressing stellate cells, it is very likely that grid cells are present also among pyramidal cells, along with other functional cell types (*Domnisoru et al., 2013*; *Sun et al., 2015*). Taken together, the observations suggest that there is no 1:1 relationship between morphological and functional cell types in layer II of the MEC. One possibility is that grid cells are specific subclasses of pyramidal and stellate cells, perhaps with a distinct molecular expression profile. In agreement with this possibility, in vitro recordings have found some heterogeneity among both stellate cells and pyramidal cells (*Fuchs et al., 2016*; *Ferrante et al., 2017*; *Giocomo et al., 2007*; *Giocomo and Hasselmo, 2009*; *Shay et al., 2016*). Alternatively, the cell class or subclass might make little difference for the functional identity of the cell. In the visual cortex, where there are also functionally defined cell types such as simple and complex cells, a cell's morphology, projection pattern, and layer assignment does not define its functional class in a straightforward manner (*Gilbert and Wiesel, 1979*). Further studies are needed to establish exactly what determines the functional identity of the cells in the MEC.

The diversity of functional cell types in the stellate population is consistent with the proposal that grid cells contribute to, but are not the sole determinant of, place cell firing (*Zhang et al., 2013*). The observation fits with earlier results that place cells show mature firing patterns before grid cells in development (*Langston et al., 2010*; *Wills et al., 2010*; *Bjerknes et al., 2018*) and place cells can maintain some significant spatial tuning even when grid cells lose their hexagonal firing patterns (*Brandon et al., 2011*; *Koenig et al., 2011*). Such residual tuning could be upheld by border cells in MEC, which are present from the earliest days that place cells have been recorded in the hippocampus (*Bjerknes et al., 2014*; *Muessig et al., 2015*), and whose spatial modulation is retained under circumstances that compromise the spatial periodicity of grid cells (*Miao et al., 2017*). These cells, as well as the recently discovered entorhinal object-vector cells (*Hoydal et al., 2018*), may provide vector-based spatial information to hippocampal place cells, as proposed by theoretical models proposing cells with such properties in the MEC or elsewhere (*O'Keefe and Burgess, 1996*; *Hartley et al., 2000*; *Burgess et al., 2000*). Whether border cells and object-vector cells are stellate cells remains to be determined, since only one of the tagged cells was a border cell in the present study. The low number of tagged border cells may reflect either that such cells are not stellate cells or that we missed the border cells, given their low abundance in MEC (5–10% [*Bjerknes et al., 2014*; *Solstad et al., 2008*; *Boccara et al., 2010*]) and the patchy organization of the MEC network (*Kitamura et al., 2014*; *Burgalossi et al., 2011*; *Heys et al., 2014*). Finally, theoretical work (*Savelli and Knierim, 2010*; *de Almeida et al., 2012*; *Rolls et al., 2006*) has suggested that place cells may also be formed by inputs from cells with weak spatial modulation in the MEC (*Zhang et al., 2013*; *Diehl et al., 2017*), or the LEC (*Hargreaves et al., 2005*), or elsewhere. By showing that grid cells account for only a fraction of the hippocampal stellate-cell input, the present findings point to multiple functional classes of MEC cells as possible sources for place-cell formation.

Taken in combination with previous recording and imaging efforts (*Domnisoru et al., 2013*; *Schmidt-Hieber and Häusser, 2013*; *Sun et al., 2015*; *Latuske et al., 2015*), our findings settle the controversy over whether stellate cells can be grid cells and suggest a major but not exclusive role for grid cells in the formation and maintenance of place cells and other spatially modulated cells in CA3 and dentate gyrus of the adult hippocampus. Taken together with recent work showing that stellate cells are required for path integration behaviors (*Tennant et al., 2018*), our data also implicate grid cells in path integration, in agreement with a large body of present and past theoretical models of grid cells (*McNaughton et al., 2006*; *Couey et al., 2013*).

# Materials and methods

## Subjects

We performed optogenetic tagging (N = 7; 5 males and 2 female) and anatomical characterization (N = 4; 2 females and 2 males) of crosses between a neuropsin-tTA ('EC-tTA') line and a tTA-dependent ArchT (tetO-ArchT-GFP) line. Both transgenic lines were bred in a C57BL6/DBA background. The generation of these two lines has been described elsewhere (*Yasuda and Mayford, 2006*; *Weible et al., 2014*). Only mice that were double positive for the two transgenes were used in this study. We also used data from three additional C57BL6/J male mice with confirmed layer II recordings for the data in *Figure 4*. All mice were between 3 and 6 months of age at time of implant.

Before implantation, mice were group housed with up to three littermates (cage size: 32cm × 17 cm × 15 cm height). Prior to surgery and testing, the mice were handled and pretrained in the recording environment at least twice. After implantation, the mice were housed individually in transparent Plexiglass cages (36cm × 24 cm×26 cm height). The mice were maintained on a 12 hr light/12 hr dark schedule and tested in the dark phase. The mice were never put on food or water restriction.

The experiments were performed in accordance with the Norwegian Animal Welfare Act and the European Convention for the Protection of Vertebrate Animals used for Experimental and Other Scientific Purposes.

## Optotrode construction

Tetrodes were constructed from four twisted 17 µm polyimide-coated platinum-iridium (90–10%) wires (California Fine Wire, CA). Four tetrodes were inserted into a 22-gauge metal cannula mounted onto a microdrive (Axona Ltd., Herts, UK). The tetrodes were cut to length using a sharp pair of scissors and a 100 micron diameter fiber with a conical tip (Doric Lenses, MFC_100/125–0.37_17 mm, ZF1.25, C45) was placed on the anterior side of the tetrode bundle. The tip of the fiber was approximately 200 microns above the tips of the tetrodes. The electrode tips were plated with platinum to reduce electrode impedances to between 150 and 300 kΩ at 1 kHz using a NanoZ device (Neuralynx, Bozeman, MT).

## Surgery

Anesthesia was induced by placing the animal in a closed glass box filled with 5% isoflurane flowing at a rate of 1.2 L/min. Afterwards, the mice were rapidly moved into the stereotaxic frame, which had a mask connected to an isoflurane pump. Air flow was kept at 1.2 L/min with 0.5–2% isoflurane as determined by physiological monitoring. Mice were then given two pain killers (Temgesic and Metacam) plus a local anesthetic (Marcain) underneath the skin over the skull. After exposing the skull, a single hole was drilled on the skull anterior to the transverse sinus to reach the entorhinal cortex. The mice were implanted with the optotrode aimed at medial entorhinal cortex. The coordinates for optotrode implants were: 3.25 mm medial-lateral relative to lambda on the left hemisphere, 0.35 mm anterior to the border of the sinus, and 0.7–0.8 mm dorso-ventral relative to the surface of the brain. The inclination of the entorhinal tetrodes was 3–4° pointing in the posterior direction. A ground wire soldered to a screw was placed in the occipital bone (over the cerebellum) and a second anchoring screw was placed in the right parietal bone. The drive was affixed to the skull using a three stage process. First, a thin layer of Optibond (Kerr, CA, USA) was applied to the skull and cured with UV light; next, a thicker layer of Charisma (Kulzer, Hanau, Germany) was applied and cured with UV light; finally, dental cement (Meliodent, Hanau, Germany) was used to bond the foot of the drive to the Charisma layer.

## Recordings and optogenetic tagging

Testing began only after a complete recovery from surgery (approximately 1 week). A session consisted of two phases. First, neural activity was recorded as the animal freely explored either a 0.8 m or 1 m black box with a single sheet of white, laminated A4 paper fixed to one wall of the box, which served as a polarizing cue. For recordings, the microdrive was connected to the recording equipment via an a.c coupled unity-gain operational amplifier and passed via wires to a digital acquisition system (Axona Ltd.). The animal was then placed into the box and allowed to move around freely

while the experimenter periodically threw crumbs of cookies into the environment. Recorded signals were amplified 5000 to 10000 times and band-pass filtered between 0.3 and 7 kHz. Triggered spikes were stored to disk at 48 kHz (50 samples per waveform, 8 bits/sample) with a 32 bit time stamp (clock rate at 96 kHz). EEG was recorded single-ended from one or more of the electrodes. The EEG was amplified 3000–10 000 times, low-pass-filtered at 500 Hz, sampled at 4800 Hz, and stored with the unit data. A tracker system (Axona Ltd.) was used to record the position of two LEDs attached to the head stage at a rate of 50 samples per second, allowing tracking of position and head direction.

In the second phase, the animal was placed into a smaller towel-lined plexiglass box while still connected to the recording equipment for optogenetic tagging of the cells. The implanted fiber was connected to a 200 mW green (532 nm) laser (Laser Century, Shanghai, China) via a patch cable (Doric Lenses, Quebec Canada). The power was approximately 5 mW from the end of the patch cable (approximately 3.5 mW from the tip of the fiber). A mechanical shutter (Uniblitz shutter system, Vincent Associates, NY, USA) was used to deliver pulses of light through the patch cable. The shutter was controlled by TTL pulses sent from an Arduino Uno (Arduino, Italy). 10–20 ms light pulses were delivered 1 to 10 times per second. A sensor in the shutter triggered a second TTL pulse that was sent to the acquisition system for synchronizing the light pulses with the electrophysiological recordings. In order to obtain the most accurate estimate of latency to inhibition, thousands of trials were run over a 30 to 60 min session. During the procedure, the small holding box was occasionally moved approximately 10 cm in order to recruit new cells that might be active in a different room location. We considered the cell to be silenced if the activity in the baseline period was more than during the laser-on period (t-test with stringent cutoff of p<0.001). The latency to inhibition was determined using a change point analysis (*Wolff et al., 2014*). The tetrodes were advanced approximately 25 microns at the end of each trial and allowed to settle overnight before the next session. This procedure was repeated until single units could no longer be isolated (10–50 recording sessions per animal, median = 25).

## Spike sorting

Spike sorting was performed offline using KlustaKwik (*Kadir et al., 2014*; *Rossant et al., 2016*). Spikes were first automatically sorted and then extensively refined manually (including elimination of poorly isolated clusters, merging of clusters, and refining of cluster boundaries) through a graphical user interface (Klustaviewa). The baseline and tagging sessions were merged and cut as one block and then split up for subsequent analysis.

## Measures used for cell type classification

All data were analyzed using custom Matlab and Python code. For circular statistics, we adapted scripts from the circstat toolbox (*Berens, 2009*).

### Gridness score (*Langston et al., 2010*; *Sargolini et al., 2006*)

The gridness score for each cell was determined from a series of expanding circular samples of the spatial autocorrelogram, each centred on the central peak but with the central peak excluded. The radius of the central peak was defined as either the first local minimum in a curve showing correlation as a function of average distance from the center, or as the first incidence where the correlation was under 0.2, whichever occurred first. The radius of the successive circular samples was increased in steps of 1 bin (2.5 cm) from a minimum of 10 cm more than the radius of the central peak, to a maximum of 90 cm. For each sample, we calculated the Pearson correlation of the ring with its rotation in $\alpha$ degrees first for angles of 60° and 120° and then for angles of 30°, 90° and 150°. We then defined the minimum difference between any of the elements in the first group (60° and 120°) and any of the elements in the second (30°, 90° and 150°). The cell's gridness score was defined as the highest minimum difference between group-1 and group-2 rotations in the entire set of successive circular samples.

### Mean vector length (head-direction score) (*Langston et al., 2010*)

Given the head-direction tuning map of a cell, if the bin $i$ with orientation $\theta_i$ expressed in radians is associated with a firing rate $\lambda_i$, the mean vector length was computed as

$$\left| \frac{\sum \lambda_i e^{i\theta_i}}{\sum \lambda_i} \right|$$

where the sums were performed over all $N$ directional bins and the modulus of the resulting complex number was obtained. A cell was considered a head direction cell if it exceeded the shuffling criterion and had an inter-trial stability of more than 0.6. The inter-trial stability was defined as the Pearson's correlation between the tuning map in the first half of the trial and the second half.

### Information per spike (*Skaggs et al., 1996*)

Given a spatial or head-direction map with mean firing rate $\lambda$ and a value $\lambda_i$ for each of its $N$ bins, information rate was computed as

$$\sum_{i=1}^{N} p_i \frac{\lambda_i}{\lambda} \log_2 \left( \frac{\lambda_i}{\lambda} \right)$$

where $p_i$ is the occupancy probability of bin $i$.

### Border score (*Solstad et al., 2008*)

The border score was computed as the difference between the maximal length of a wall touching on any single firing field of the cell and the average distance of the field from the nearest wall, divided by the sum of those values. The range of border scores was thus $-1$ to $1$. Firing fields were defined as collections of neighboring pixels with firing rates higher than 20% of the cell's peak firing rate and a size of at least 200 cm$^2$.

### Speed score (*Kropff et al., 2015*)

The speed score for each cell was defined as the Pearson product-moment correlation between the cell's instantaneous firing rate and the rat's instantaneous running speed, on a scale from $-1$ to $1$.

### Shuffling

A cell was defined as a functional cell type if its score in a functional category exceeded a chance level determined by repeated shuffling of the experimental data (200 permutations per cell). For each permutation, the entire sequence of spikes fired by the cell was time-shifted along the animal's path by a random interval between $\pm$ 20 s, with the end of the session wrapped to the beginning. Time shifts varied randomly between permutations and between cells. From the shuffled distribution, we calculated the 95th percentile and used this value as a threshold for assigning cells into a particular functional class.

### Classification of cells into putative stellate and putative pyramidal using the *Tang et al. (2014)* classifier

The local field potential was band-pass filtered (4–12 Hz). The Hilbert transform was then used to determine the instantaneous phase of the theta oscillation. The strength of locking to theta phase and the preferred phase angle was determined by the Rayleigh vector. These two properties were then used to classify each cell as a putative pyramidal or putative stellate using the code published in the *Tang et al. (2014)* study. Cells within 0.1 of the decision boundary were placed in the 'guard zone.' We also clustered the cells using an agglomerative clustering method with the number of clusters set to 2 (*Berens, 2009*).

### Statistical tests and data availability

All comparisons were two sided. Due to the non-normal distributions of the measures used for cell type classification (see violin plots in *Figures 3* and *4*), we used nonparametric statistics for those comparisons. For analysis of the cell's firing relationship to theta phase, we used circular statistics (*Berens, 2009*).

Python code and preprocessed source data used for statistical analysis and visualization relating to *Figures 3* and *4* are available on GitHub (https://github.com/davidcrowland/

archdata, *Rowland, 2018a*; copy archived at https://github.com/elifesciences-publications/arch-data). Unprocessed data are archived on Norstore (https://archive.norstore.no/) (*Rowland, 2018b*).

## In situ hybridization and antibody staining

Mice were perfused transcardially with 4% paraformaldehyde (PFA) in PBS. The brain was extracted and stored in 4% PFA overnight before being transferred to 30% sucrose solution for approximately 2 days. The brain was then sectioned sagitally in 30 micrometer thick sections and divided into a set of approximately 6 series and stored in a −80°C freezer. A series was then thawed before use.

To stain for reelin and ArchT, 30 µm thaw-mounted sections were hybridized overnight at 62°C with a DIG-labeled riboprobe for reelin (1:400; Roche, Cat. 11277073910) and FITC-labeled ribo-probe for Arch (1:500; Roche, Cat. 11685619910) and then incubated at room temperature for 4 hr in the blocking solution (0.1M TRIS-HCL pH7.5, 0.15 M NaCL, 0.5% Blocking reagent Perkin Elmer TSA kit). Next, the anti-Fluor-HRP (1:1000; Invitrogen, Cat. A21253) antibody was added and the sections were incubated overnight at room temperature. The tissue was washed with TBST buffer and the Fluorescein signal was developed for 45 min at room temperature using TSA Plus Fluorescein kit (1:50; PerkinElmer, Cat. NEL7410001KT). The sections were then placed in the blocking solution for 4 hr before incubated overnight at 4°C in Anti-Digoxigenin-POD (1:1000; Roche, Cat. 11207733910). The tissue was exposed to TSA Plus Cyanine 3 (1:50; PerkinElmer, Cat. NEL744001KT) for 45 min at room temperature, then washed in TBST buffer.

For calbindin antibody and ArchT in situ staining, we first performed the in situ for ArchT as above but with a dilution of 1:600 of the DIG-labeled riboprobe. We then proceeded with the anti-body stain for calbindin. The sections were washed twice with an incubation solution of PBS + 0.3% Triton + 3% BSA and then incubated for 24 hr at room temperature with the primary antibody diluted in a solution of PBS + 0.3% Triton + 3% BSA. The following day, the sections were washed three times with a solution of PBS + 0.1% Triton+1% BSA and incubated overnight (17.5 hr) at room temperature in the secondary antibody in a solution of PBS + 0.1% Triton + 1% BSA. We used a dilu-tion of 1:1000 for the primary antibody (Monoclonal anti-Calbindin D-28k, Swant) and 1:700 for the secondary (Donkey anti-Rabbit Cy3; Jackson Immuno Research, 711-165-152).

## Confocal microscopy

Sections were scanned as a stack of images under a confocal microscope (Zeiss LSM800, Zeiss, Ger-many) using a 40x or 63x oil immersion objective at multiple planes in the z dimension. The amount of overlap in the MEC was quantified manually in Imaris (Bitplane, Zurich, Switzerland). For presenta-tion (*Figure 1* and *Figure 1—figure supplement 1*), the resulting Z-stacks were then collapsed into a maximum intensity projection and pseudo-colored as cyan and magenta for display (ImageJ; Adobe Photoshop, Adobe Systems Inc. CA).

## Acknowledgements

The work was supported by two Advanced Investigator Grants from the European Research Council (GRIDCODE', Grant Agreement N°338865, to EIM; ENSEMBLE', Grant Agreement N°268598, to M-BM), a NEVRONOR grant from the Research Council of Norway (grant no. 226003 to EIM), the Centre of Excellence scheme and the National Infrastructure Scheme of the Research Council of Nor-way (Centre for Neural Computation, grant number 223262; NORBRAIN1, grant number 197467), the Louis Jeantet Prize, the Körber Prize, the Kavli Foundation (all to M-BM and EIM). and an IIF grant from the Marie Sklodowska-Curie Actions to DCR. We thank K Haugen, H Waade, and V Frolov for technical assistance, and Flavio Donato and Øyvind A Høydal for additional data which contributed to *Figure 4*.

## Additional information

### Funding

| Funder | Grant reference number | Author |
| --- | --- | --- |
| H2020 European Research Council | | Edvard I Moser<br>May-Britt Moser |

| Norges Forskningsråd | | Edvard I Moser<br>May-Britt Moser |
| --- | --- | --- |
| Louis-Jeantet Foundation | | Edvard I Moser<br>May-Britt Moser |
| Körber-Stiftung | The Körber prize | Edvard I Moser<br>May-Britt Moser |
| Kavli Foundation | | Edvard I Moser<br>May-Britt Moser |
| H2020 Marie Skłodowska-Curie Actions | | David Clayton Rowland |

The funders had no role in study design, data collection and interpretation, or the decision to submit the work for publication.

### Author contributions

David C Rowland, Conceptualization, Software, Funding acquisition, Investigation, Methodology, Writing—original draft, Writing—review and editing; Horst A Obenhaus, Resources, Software, Formal analysis, Validation, Investigation, Methodology, Writing—review and editing; Emilie R Skytøen, Qiangwei Zhang, Investigation, Methodology, Writing—review and editing; Cliff G Kentros, Supervision, Methodology, Writing—review and editing; Edvard I Moser, Conceptualization, Resources, Supervision, Funding acquisition, Investigation, Writing—original draft, Project administration, Writing—review and editing; May-Britt Moser, Resources, Supervision, Funding acquisition, Writing—original draft, Project administration, Writing—review and editing

### Author ORCIDs

David C Rowland http://orcid.org/0000-0002-2735-2730

### Ethics

Animal experimentation: The experiments were performed in accordance with the Norwegian Animal Welfare Act and the European Convention for the Protection of Vertebrate Animals used for Experimental and Other Scientific Purposes. Norwegian Food Safety Authority FOTS ID 7163.

### Decision letter and Author response

Decision letter https://doi.org/10.7554/eLife.36664.018
Author response https://doi.org/10.7554/eLife.36664.019

## Additional files

### Supplementary files

• Transparent reporting form
DOI: https://doi.org/10.7554/eLife.36664.014

### Data availability

Data and python code required to reproduce Figures 3 and 4 are available on GitHub (https://github.com/davidcrowland/archdata; copy archived at https://github.com/elifesciences-publications/archdata). Unprocessed electrophysiological data are archived on Norstor (https://archive.sigma2.no/pages/public/datasetDetail.jsf?id=10.11582/2018.00025).

The following dataset was generated:

| Author(s) | Year | Dataset title | Dataset URL | Database, license, and accessibility information |
| --- | --- | --- | --- | --- |
| David Clayton Rowland | 2018 | Data from Rowland et al 2018 | http://dx.doi.org/10.11582/2018.00025 | Publicly available at the NIRD Research Data Archive (DOI: http://dx.doi.org/10. |

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
