## [Decision Letter]

Thank you for submitting your article "Functional properties of stellate cells in medial entorhinal cortex layer II" for consideration by *eLife*. Your article has been reviewed by Michael Frank as the Senior Editor, a Reviewing Editor, and three reviewers. The following individual involved in review of your submission has agreed to reveal his identity: Michael Hasselmo (Reviewer #1).

The reviewers have discussed the reviews with one another and the Reviewing Editor has drafted this decision to help you prepare a revised submission.

Summary:

The reviewers agree that the authors present important results using optogenetic tagging of stellate cells in the medial entorhinal cortex. Using this technique, they show that almost one-fourth of the neurons tagged as stellate cells are identified as grid cells. Next, they test the classifier used by Tang et al., to classify cells as putative pyramidal and putative stellate cells. Among a larger population of cells, they find that putative stellate cells are more likely than putative pyramidal cells to be grid cells (32.4% vs. 22.6%). Moreover, the theta rhythmicity of cells is a poor predictor of grid cell activity, and the classifier is an unreliable means of identifying the subpopulation of stellate cells that are labeled by the transgenic mouse line used in this study. The reviewers agree that this is an important fundamental contribution to the literature that resolves a controversy concerning the morphological properties of grid cells.

Although a number of criticisms were raised, the reviewers agree that these should be easily addressable with minor analyses, wording changes, and clarification of cell numbers and percentages. A summary of the reviewers' points is provided below.

Essential revisions:

1) The wording and presentation of the main result seems misleading. With optotagging, the authors find that 23.2% of tagged stellate cells are grid cells, but they find that 33.8% of the untagged population (pyramidal cells and unlabeled stellate cells) are grid cells. Assuming the untagged stellate cells contain a similar fraction of grid cells as the tagged stellate cells, then this implies that in fact a larger fraction of pyramidal cells are grid cells compared to stellate. This is counter to the conclusion that the authors cite from the Tang classifier (see for example the Introduction). The authors should explicitly address the discrepancy here and provide the reader with guidance on whether grid cells are more likely to be stellate or pyramidal cells. The Introduction paints the picture that stellate cells form the larger fraction of grid cells relative to pyramidal cells. Indeed, the authors say: "large fraction of stellates are grids". The untagged population had a larger (34%) fraction of grids than the tagged stellates (23%). So, 23% only appears large compared to the 3/94 estimate from the Tang paper. The phrase "substantial fraction" instead of "large fraction" would be better. The authors should more clearly layout the findings from each method so that the readers do not misinterpret the findings here.

2) Related to the above point, the authors found that many putative stellate cells were grid cells (~33%) using the classifier of Tang et al. Why did the authors get such a different result compared to Tang et al? This point should be explicitly addressed. Moreover, Figure 4—figure supplement 1 indicates that the Tang et al. classifier is not effective. This result diminishes the impact of the data shown in Figure 4. Figure 4—figure supplement 1 could be shown at the beginning of main Figure 4 to increase the awareness of this issue. The demonstration that classifier is not accurate makes its use very limited.

3) The numbers and fractions of cells that the authors report seem inconsistent. In the Results section, the fraction of cells making up the mixed populations do not make sense. How can 8.6% of cells be mixed grid cells when the mixed populations of other types only add up to 4.9%? Also, the numbers of cells they report seems inconsistent. They mention in line 108 that the population of recorded cells from the EC-tTA mice is 611 and that they tagged 82. Using the percentages they quote- 33.8% grid cells in the untagged and 23.2% in the tagged population gives ~198 grid cells (0.338*(611-82)+0.232*82=198). However, in the Results section they say that there were 411 grid cells from the EC-tTA mice. Later in Figure 4—figure supplement 1, they mention tagged cells from a population of 75 as opposed to 82.

4) The authors show a few nice examples of optotagged cells (Figure 2C, E). Presumably these are some of the best examples. It would be useful for others wishing to use this technique to also see examples of cells that just barely pass the optotagging inclusion criteria (described in subsection “Recordings and optogenetic tagging”).

5) Only 32% of stellate cells were labeled by the EC-tTA line. Is this a random selection among all stellate cells, or do these labeled cells represent a distinct subpopulation of stellate cells with potential functional differences?

6) The results for linking grid cell properties to theta rhythms are not explained clearly. The authors suggest that theta phase preference is not a good indication of a grid cell, but this is misleading as they record from many grid cells with strong theta modulation. They should note that many grid cells are strongly theta rhythmic, and only that the distribution of theta rhythmic neurons is slightly stronger in the non-tagged population. Also, the authors suggest that the diversity in the preferred firing and depth of modulation in Figure 4A, c is because of the propensity of grid cells to show theta phase precession. However, wouldn't precession push theta phase preference toward zero and create a lower depth of modulation? Perhaps they could directly show examples of precession affecting the spread of the dots in Figure 4. Also, it is somewhat misleading that the authors emphasize that the grid score negatively correlates with the depth of theta modulation (Figure 4—figure supplement 3), but show that stellate cells with large grid scores still have the same theta phase vs. theta modulation distribution (Figure 4—figure supplement 1). They should reduce the emphasis on the negative correlation.

7) In their conclusions, the authors state that their results suggest a major role for grid cells in the maintenance and formation of place cells- but this does not seem consistent with their results. They don't show that these stellate cells project to hippocampus. Stating in the conclusion that stellate grid cells play a major role in place field formation is not fully justified in the context of this study. Additional work is needed to more strongly confirm this possibility or the authors should be clearer in their conclusions about the limitations of their results. If the authors choose to keep this strong conclusion in the paper, one important piece of histological data that would bolster this conclusion would be to include images of the projections from ArchT expressing cells entering the middle molecular layer of the dentate and stratum lacunosum moleculare of CA3.

8) Figure 3B: Is a Mann-Whitney U-test appropriate given that the majority of values for grid score fall below 0 for both populations?

9) Accidental resampling of cells is a potential serious concern. The number of recorded neurons seems very high when considering the methodology, the histology and the difficulty of recording large populations of neurons from MEC. The authors state in the paper their tetrodes can record from ~100μm of tissue – yet they also say that they move their tetrodes 25μm each day. Surely this should be a larger distance to avoid resampling. This is a potentially a significant issue when doing a study which is attempting to count neurons and calculate percentages of a population with certain functional tuning. The authors should supply information about the number of sessions that were recorded from each animal at a bare minimum. The histology in Figure 1—figure supplement 3 shows the final recording location for each animal and supports the possibility that there is some resampling going on here as the authors have not been able to drive the tetrode bundle very far through the dorsal to ventral axis of MEC. The angle of the implant is not sufficiently matched the angle of the cell layer to allow for a large number of unique recordings to be taken whilst the tetrode bundle remains in layer 2. Some possible methods to test for whether resampling may be occurring would be to see if there is a higher than chance level of correlation between numbers of functional cell types or optogenetically responsive cells on a tetrode from day to day. Another possibility would be to take only every fifth session from each mouse when the tetrodes should have travelled 100μm and repeat the counts with that data to see if the percentages are comparable.

---

## [Author Response]

Essential revisions:1) The wording and presentation of the main result seems misleading. With optotagging, the authors find that 23.2% of tagged stellate cells are grid cells, but they find that 33.8% of the untagged population (pyramidal cells and unlabeled stellate cells) are grid cells. Assuming the untagged stellate cells contain a similar fraction of grid cells as the tagged stellate cells, then this implies that in fact a larger fraction of pyramidal cells are grid cells compared to stellate. This is counter to the conclusion that the authors cite from the Tang classifier (see for example the Introduction). The authors should explicitly address the discrepancy here and provide the reader with guidance on whether grid cells are more likely to be stellate or pyramidal cells. The Introduction paints the picture that stellate cells form the larger fraction of grid cells relative to pyramidal cells. Indeed, the authors say: "large fraction of stellates are grids". The untagged population had a larger (34%) fraction of grids than the tagged stellates (23%). So, 23% only appears large compared to the 3/94 estimate from the Tang paper. The phrase "substantial fraction" instead of "large fraction" would be better. The authors should more clearly layout the findings from each method so that the readers do not misinterpret the findings here.

We understand the reviewers’ point and have modified the text accordingly. Specifically, we have changed “large” to “substantial.” We have also removed the discussion of “putative stellate” and “putative pyramidal” from the Introduction as it unnecessarily confuses the issue at this stage. Rather we address the key point that the classifier does not partition grid cells and non-grid cells into separate categories. In the introduction, we now write:

“Consistent with prior observations ^25^, we found that neither the classifier’s cell assignment nor the relationship between the firing of the cell and the theta oscillation cleanly separated grid cells from other cell types.”

And:

“Taken together, the results suggest that substantial fraction of the stellate cells are grid cells, though they can have other functional identities as well, and the relationship between the cell’s firing and the theta oscillation is a poor predictor of the functional identity the cell.”

We believe that these changes together with the final sentence in the abstract and second paragraph in the discussion make the point clear that only a fraction of stellate cells are grid cells, but that the observed number (~25%) is large enough to substantially impact place cell firing. This fits with the existing literature that grid cells contribute to but are not the sole determinant of place cell firing.

As far as laying out the findings more clearly, we have followed the reviewers’ later suggestion and put the classifier performance on the main figure 4 and addressed that first in the main text. This helps to establish the limitations of the approach for identifying stellate and pyramidal cells. Because their classifier also partitioned grid cells, we believed that it was worth moving ahead with the classifier and evaluating how well it did at capturing grid cells. We show that the classifier does not separate out grid cells in our dataset.

2) Related to the above point, the authors found that many putative stellate cells were grid cells (~33%) using the classifier of Tang et al. Why did the authors get such a different result compared to Tang et al? This point should be explicitly addressed. Moreover, Figure 4—figure supplement 1 indicates that the Tang et al. classifier is not effective. This result diminishes the impact of the data shown in Figure 4. Figure 4—figure supplement 1could be shown at the beginning of main Figure 4 to increase the awareness of this issue. The demonstration that classifier is not accurate makes its use very limited.

We puzzled over this issue ourselves, but we still do not have a clear answer. One obvious point is that the fraction of grid cells in the Tang et al., population is very low for layer II (approximately 10%). In mice, the estimates for numbers of grid cells range from 20-30% with a 95^th^ percentile cutoff (this study; Heys et al., 2014; Latuske et al., 2014) and >15% with a 99^th^ percentile when layers II/III are combined (Giocomo et al., 2011). In rats, the species tested by Tang et al., studies have noted grid cell prevalence in layer II range from 25 to 50% (Boccara et al., 2010; Kropff et al., 2015). We have tried to match our statistics to the Tang et al. study as best we can (e.g. we also used a 95^th^ percentile cutoff for assigning cells into a particular category and used the same statistical tests), so we are reasonably sure that it is not an artifact of the way we are assigning cells into the grid category or comparing populations. Moreover, the Latuske et al. study used the same classifier and got similar results as we did using a dataset that was roughly 20% grid cells (of 302 putative layer II cells). Notably, the Sun et al., study, which used 1-photon miniscopes, found approximately 10% of cells (in both the stellate and pyramidal cell population) were grid cells, but this method likely underestimates the numbers of grid cells. GCAMP imaging will only capture the highest firing rate cells and some cells with narrow spacing could be lost due to smearing out of the fields caused by the broad calcium transients. Thus, the numbers of grid cells in the Tang et al., study is below what is expected for layer II recordings.

Besides the statistical characterization, the example grid cells in Tang et al. Figure 2 are not the most convincing examples and 2 of the 3 example cells have a grid spacing that barely fits within the box. It is therefore possible that the discrepancy in the number of grid cells is due to the differences in tetrode positioning in the Tang study compared to ours (e.g. they had more ventral recording sites), but the authors did not provide any images of the histology, making it impossible to compare ours to theirs. The lack of histology in their paper also precludes any examination of whether the recordings might have included cells from the deeper layers in their analysis.

Finally, there is some evidence of anatomical clustering of grid cells (see Heys et al., 2014) and by extension other functional cell types, which might have contributed to the different results. Taken together, we believe that our dataset (and the others mentioned above) provide a more complete and accurate picture of the diversity of cells in layer II of the MEC. The larger numbers of animals and cells likely smoothed out the effects of clustering and variations in tetrode locations. Ultimately, we might still need more advanced methods (such as two-photon imaging with fast sensors and an imaging window that covers large portions of the MEC) in order to have a precise estimate of the fractions of cells in each category. However, our estimates appear to compare favorably with estimates made in earlier studies and are likely representative of layer II.

We have put a condensed version of this explanation in the discussion. We have also followed the reviewers’ suggestion and now begin Figure 4 with a stacked bar graph showing how well the classifier worked on our tagged population (as discussed in our response to point 1).

3) The numbers and fractions of cells that the authors report seem inconsistent. In the Results section, the fraction of cells making up the mixed populations do not make sense. How can 8.6% of cells be mixed grid cells when the mixed populations of other types only add up to 4.9%? Also, the numbers of cells they report seems inconsistent. They mention in line 108 that the population of recorded cells from the EC-tTA mice is 611 and that they tagged 82. Using the percentages they quote- 33.8% grid cells in the untagged and 23.2% in the tagged population gives ~198 grid cells (0.338*(611-82)+0.232*82=198). However, in the Results section they say that there were 411 grid cells from the EC-tTA mice. Later in Figure 4—figure supplement 1, they mention tagged cells from a population of 75 as opposed to 82.

We are grateful to the reviewers for pointing out this important issue. Just as a point of clarification, though, the 411 grid cells are from the “extended dataset”, which includes three additional animals and roughly two times as much data. The Tang et al., classifier only needs layer II recordings that are confirmed by histology, so we were able to include more cells from non-transgenic mice. To make this point clearer, we have split one long sentence into two in main text. It now reads:

“We therefore evaluated their classifier on an extended dataset of 1332 cells including 411 grid cells. This extended dataset included recordings from the EC-tTA x tetO-ArchT-GFP mice and three additional mice with histologically confirmed layer II recordings (N = 270 putative pyramidal cells, 999 putative stellate and 63 cells in the guard zone).”

Nevertheless, the reviewers’ comment that the percentages do not quite add up exposed a minor but important bug in our analysis. In particular, some “NaNs” which should have been dropped were not which caused small (<2.5% overall) changes to percentages. We have updated and checked the text that all of the numbers sample sizes and percentages match. Dropping the “NaNs” also changed the outcome of one important statistical test: the grid score between the tagged and untagged population came out significant after removing the “NaNs.” As the reviewers mention in point 1, this was already evident in the percentages of grid cells in the two populations and is now even more strongly written into the text.

4) The authors show a few nice examples of optotagged cells (Figure 2C,E). Presumably these are some of the best examples. It would be useful for others wishing to use this technique to also see examples of cells that just barely pass the optotagging inclusion criteria (described in subsection “Recordings and optogenetic tagging”).

We have added additional examples into new Figure 2—figure supplement 1. These cells were pulled from the bottom 30% of the inhibited population but nevertheless show clear inhibition to the light.

5) Only 32% of stellate cells were labeled by the EC-tTA line. Is this a random selection among all stellate cells, or do these labeled cells represent a distinct subpopulation of stellate cells with potential functional differences?

This is an interesting question but one that we do not know the answer to at the moment. As the reviewers imply, and we write in the text, there is some evidence of different types of stellate cells based on morphology (e.g. intermediate stellate cells) or electrophysiological properties in vitro. Unfortunately, there are no known markers for different subclasses of stellate cells and the electrophysiological and morphological criteria are not clear-cut. Therefore, even additional experiments might not reveal whether the cells belong to a particular subclass. We have added additional text explicitly stating this potential limitation to the study as follows:

“The fact that not all stellate cells express Arch raises the possibility that the EC-tTA line targets a subclass of stellate cells, but we could not test for that because there are no known markers for any subclass.”

6) The results for linking grid cell properties to theta rhythms are not explained clearly. The authors suggest that theta phase preference is not a good indication of a grid cell, but this is misleading as they record from many grid cells with strong theta modulation. They should note that many grid cells are strongly theta rhythmic, and only that the distribution of theta rhythmic neurons is slightly stronger in the non-tagged population. Also, the authors suggest that the diversity in the preferred firing and depth of modulation in Figure 4A, c is because of the propensity of grid cells to show theta phase precession. However, wouldn't precession push theta phase preference toward zero and create a lower depth of modulation? Perhaps they could directly show examples of precession affecting the spread of the dots in Figure 4. Also, it is somewhat misleading that the authors emphasize that the grid score negatively correlates with the depth of theta modulation (Figure 4—figure supplement 3), but show that stellate cells with large grid scores still have the same theta phase vs. theta modulation distribution (Figure 4—figure supplement 1). They should reduce the emphasis on the negative correlation.

We completely agree with the reviewers’ points here. We do not wish to say that grid cells are not theta modulated, nor do we wish to leave readers with the impression that all grid cells have less theta modulation than non-grid cells. Indeed, we try to make the point that although there are some significant differences in our population, the trends are weak (theta modulation explains only approximately 2% of the variance in grid score) and nearly all cells across functional categories are theta modulated. This leads to our main conclusion that theta modulation is a poor predictor of the functional identity of the cell (i.e. there is strong theta modulation in the population as a whole with considerable overlap in the depth of modulation across functional cell types).

This is explicitly addressed in the Results section as follows:

“In summary, theta modulation is prevalent across all functional cell types in the MEC but with some small, though significant, differences between groups. For example, we observed a small negative correlation between grid score and depth of theta modulation but the correlation accounted for approximately 2% of the variance. These results show that, despite some differences between groups, the large diversity in theta modulation and preferred theta phase within cell types makes the relationship between the cell and the theta oscillation a poor predictor of the functional identity of the cell, in close agreement with the results of Latuske et al., (2015)”.

Regarding the issue of phase precession and the scatter of the points in Figure 4, the essential point that phase precession will negatively affect depth of theta modulation is clearly present in the Hafting et al., 2008 paper (Figure 1 and 2). The top grid cell has clear phase precession (raster on bottom) and moderate theta modulation (histogram top center). The bottom grid cell has no phase precession (it locks to approximately 180 deg) and has stronger theta modulation. Notably, both cells have clear theta modulation, which supports the reviewers’ point. We hope that our revisions have made our position clear as well.

7) In their conclusions, the authors state that their results suggest a major role for grid cells in the maintenance and formation of place cells- but this does not seem consistent with their results. They don't show that these stellate cells project to hippocampus. Stating in the conclusion that stellate grid cells play a major role in place field formation is not fully justified in the context of this study. Additional work is needed to more strongly confirm this possibility, or the authors should be clearer in their conclusions about the limitations of their results. If the authors choose to keep this strong conclusion in the paper, one important piece of histological data that would bolster this conclusion would be to include images of the projections from ArchT expressing cells entering the middle molecular layer of the dentate and stratum lacunosum moleculare of CA3.

We now provide confirmation that arch-expressing cells project to the dentate gyrus and CA3 as Figure 1—figure supplement 2, which confirms previous studies that used the same tTa driver line (we have modified the text in the first paragraph of the results to describe this result as well). This provides proof-of-concept that the grid cells in the stellate population can influence place cell firing. In addition, one of the fascinating properties of stellate cells is that all or nearly all of them project to the ipsilateral dentate gyrus (98% of reelin positive cells project to the ipsilateral dentate according to Varga et al.,). The Kitamara et al., 2014 paper also quantified overlap between reelin and dentate projecting cells. Although they do not report raw numbers, the bar graphs clearly indicate that more than 90% of the cells reelin positive cells also projected to the hippocampus. It is always difficult to talk in absolutes when it comes to anatomy, but given that no tracer is 100% effective, it is very plausible that every stellate cell projects to the hippocampus. We believe that these two pieces of evidence provide sufficient support for our claim that grid cells can influence place cell firing, though as we note in the abstract, and discussion, they are certainly not the sole determinant of place cell firing. We have modified the text in the introduction to incorporate this result from Varga et al.:

“Nearly every stellate cell projects to the DG, CA3 and/or CA2 regions of the hippocampus, and stellate cells make up the main and nearly exclusive excitatory input from the MEC to these areas.”

8) Figure 3B: Is a Mann-Whitney U-test appropriate given that the majority of values for grid score fall below 0 for both populations?

We clearly needed a non-parametric test for this analysis because the scores are not normally distributed. The key assumption for the Mann-Whitney U-test is that the scores can be ranked, which is true for negative and positive numbers. To control for that, we added 1 to all the values and re-ran the test and the results were the same. We also tried other non-parametric test and reached the same conclusion. Perhaps the reviewers are puzzled about why the percentages and plots appear different between the groups but the test was non-significant. As we note in point 3 above, the results of this changed after removing the “nan” values and there is now a significant difference between the groups.

9) Accidental resampling of cells is a potential serious concern. The number of recorded neurons seems very high when considering the methodology, the histology and the difficulty of recording large populations of neurons from MEC. The authors state in the paper their tetrodes can record from ~100μm of tissue – yet they also say that they move their tetrodes 25μm each day. Surely this should be a larger distance to avoid resampling. This is a potentially a significant issue when doing a study which is attempting to count neurons and calculate percentages of a population with certain functional tuning. The authors should supply information about the number of sessions that were recorded from each animal at a bare minimum. The histology in Figure 1—figure supplement 3 shows the final recording location for each animal and supports the possibility that there is some resampling going on here as the authors have not been able to drive the tetrode bundle very far through the dorsal to ventral axis of MEC. The angle of the implant is not sufficiently matched the angle of the cell layer to allow for a large number of unique recordings to be taken whilst the tetrode bundle remains in layer 2. Some possible methods to test for whether resampling may be occurring would be to see if there is a higher than chance level of correlation between numbers of functional cell types or optogenetically responsive cells on a tetrode from day to day. Another possibility would be to take only every fifth session from each mouse when the tetrodes should have travelled 100μm and repeat the counts with that data to see if the percentages are comparable.

We have addressed this using the final method the reviewers proposed. Because stellate cells are not randomly dispersed in layer II (there are essentially no stellate cells in the CB+ clusters) and functional cell types show some clustering, we had a priori reasons to expect that there might be day-to-day fluctuations in numbers of functional cell types or inhibited cells. We therefore took the recorded cells and downsampled each animal’s data and recalculated the percentages. The rationale being that the percentages of cells in each functional class should be robust to downsampling of the dataset. We examined cells from the overall population recorded at least 1, 2, 3, 4 and 5 and found very similar percentages regardless of the amount we downsampled. We have included the results from 1 day (full dataset) and 4+ days apart as a new figure supplement (Figure 3—figure supplement 1) and have modified the text as follows:

“To control for the possibility that accidental resampling of cells between sessions created spurious results, we downsampled our data to cells recorded 4 or more days apart, when the tetrodes had been turned approximately 100 microns. The downsampled population had similar proportions in each functional class as the full dataset, suggesting that percentages described here are representative of the functional diversity in layer II of the MEC (Figure 3—figure supplement 1).”